# Clozapine Worsens Glucose Intolerance, Nonalcoholic Fatty Liver Disease, Kidney Damage, and Retinal Injury and Increases Renal Reactive Oxygen Species Production and Chromium Loss in Obese Mice

**DOI:** 10.3390/ijms22136680

**Published:** 2021-06-22

**Authors:** Geng-Ruei Chang, Hsien-Yueh Liu, Wei-Cheng Yang, Chao-Min Wang, Ching-Fen Wu, Jen-Wei Lin, Wei-Li Lin, Yu-Chen Wang, Tzu-Chun Lin, Huei-Jyuan Liao, Po-Hsun Hou, Chee-Hong Chan, Chuen-Fu Lin

**Affiliations:** 1Department of Veterinary Medicine, National Chiayi University, 580 Xinmin Road, Chiayi 600023, Taiwan; grchang@mail.ncyu.edu.tw (G.-R.C.); leowang@mail.ncyu.edu.tw (C.-M.W.); cfwu@mail.ncyu.edu.tw (C.-F.W.); lin890090@gmail.com (T.-C.L.); pipi324615@gmail.com (H.-J.L.); 2Bachelor Degree Program in Animal Healthcare, Hungkuang University, 6 Section, 1018 Taiwan Boulevard, Shalu District, Taichung 433304, Taiwan; lhy_vet@hk.edu.tw (H.-Y.L.); jenweilin@hk.edu.tw (J.-W.L.); ivory-lily99@gmail.com (W.-L.L.); 3School of Veterinary Medicine, National Taiwan University, 4 Section, 1 Roosevelt Road, Taipei 100046, Taiwan; yangweicheng@ntu.edu.tw; 4General Education Center, Chaoyang University of Technology, 168 Jifeng Eastern Road, Taichung 413310, Taiwan; 5Division of Cardiology, Asia University Hospital, 222 Fuxin Road, Wufeng District, Taichung 413505, Taiwan; richard925068@gmail.com; 6Department of Medical Laboratory Science and Biotechnology, Asia University, 500 Lioufeng Road, Wufeng District, Taichung 413305, Taiwan; 7Division of Cardiovascular Medicine, China Medical University Hospital, 2 Yude Road, North District, Taichung 404332, Taiwan; 8College of Medicine, China Medical University, 91 Hsueh-Shih Road, North District, Taichung 404333, Taiwan; 9Department of Psychiatry, Taichung Veterans General Hospital, 4 Section, 1650 Taiwan Boulevard, Taichung 407219, Taiwan; 10Faculty of Medicine, National Yang Ming Chiao Tung University, 2 Section, 155 Linong Street, Beitou District, Taipei 112304, Taiwan; 11Division of Nephrology, Chang Bing Show Chwan Memorial Hospital, 6 Lugong Road, Lukang Township, Changhua 505029, Taiwan; 12Department of Veterinary Medicine, College of Veterinary Medicine, National Pingtung University of Science and Technology, 1 Shuefu Road, Neipu, Pingtung 912301, Taiwan

**Keywords:** chromium, clozapine, fatty liver disease, glucose intolerance, obesity, renal damage, retinal injury, reactive oxygen species

## Abstract

Clozapine is widely employed in the treatment of schizophrenia. Compared with that of atypical first-generation antipsychotics, atypical second-generation antipsychotics such as clozapine have less severe side effects and may positively affect obesity and blood glucose level. However, no systematic study of clozapine’s adverse metabolic effects—such as changes in kidney and liver function, body weight, glucose and triglyceride levels, and retinopathy—was conducted. This research investigated how clozapine affects weight, the bodily distribution of chromium, liver damage, fatty liver scores, glucose homeostasis, renal impairment, and retinopathy in mice fed a high fat diet (HFD). We discovered that obese mice treated with clozapine gained more weight and had greater kidney, liver, and retroperitoneal and epididymal fat pad masses; higher daily food efficiency; higher serum or hepatic triglyceride, aspartate aminotransferase, alanine aminotransferase, blood urea nitrogen, and creatinine levels; and higher hepatic lipid regulation marker expression than did the HFD-fed control mice. Furthermore, the clozapine group mice exhibited insulin resistance, poorer insulin sensitivity, greater glucose intolerance, and less Akt phosphorylation; their GLUT4 expression was lower, they had renal damage, more reactive oxygen species, and IL-1 expression, and, finally, their levels of antioxidative enzymes (superoxide dismutase, glutathione peroxidase, and catalase) were lower. Moreover, clozapine reduced the thickness of retinal cell layers and increased *iNOS* and *NF-κB* expression; a net negative chromium balance occurred because more chromium was excreted through urine, and this influenced chromium mobilization, which did not help overcome the hyperglycemia. Our clozapine group had considerably higher fatty liver scores, which was supported by the findings of lowered adiponectin protein levels and increased FASN protein, PNPLA3 protein, *FABP4* mRNA, and *SREBP1* mRNA levels. We conclude that clozapine can worsen nonalcoholic fatty liver disease, diabetes, and kidney and retinal injury. Therefore, long-term administration of clozapine warrants higher attention.

## 1. Introduction

Clozapine is known to constitute a second-generation antipsychotic (SGA; also known as an atypical antipsychotic) employed in cases of schizoaffective disorder and schizophrenia in which treatment was previously unsuccessful [1]. Most crucially, compared with that of conventional neuroleptics, it is more efficacious and less likely to induce tardive dyskinesia and extrapyramidal side effects [1,2]. Among those for whom typical antipsychotics did not have an effect, 30–60% respond positively to clozapine [2]. One study discovered a genetic overlap between the pathogenesis of schizophrenia and the mechanism of clozapine as an antipsychotic, indicating that if the mechanism of clozapine could be understood, valid therapeutic targets could be identified, and perhaps, schizophrenia’s complex pathophysiology could be elucidated [3]. Researchers, however, reported associations between clozapine and adverse metabolic events—such as elevated cholesterol, blood glucose concentration, triglyceride levels, and body weight—and these events negatively influence medication compliance and health [4,5].

Clozapine achieves a favorable balance between the metabolism of serotonin and dopamine, with more positive outcomes achieved when the level of serotonin metabolism is higher than that of dopamine [6]. Moreover, clozapine causes a decrease in serotonin antagonism to induce an upregulation in the serotonin receptor and an increase in the extracellular levels of serotonin [7,8]. However, clozapine has severe and lethal side effects, such as serotonin syndrome [9]. Another critical factor is that SGAs were reported to exert significant metabolic side effects, such as elevated type 2 diabetes mellitus (T2DM) risk [10]. A study discovered that the relative diabetes risk was higher under clozapine treatment than under treatment with a conventional antipsychotic [11]. The glucose dysregulation risk was considered by one study to be elevated due to antipsychotic-drug-induced weight gain [12]. The clozapine group was twice as likely to have diabetes mellitus or impaired glucose tolerance as the group taking other antipsychotics in a cross-sectional study [13]. In addition, clozapine was associated with significantly higher glucose levels than typical antipsychotics in oral glucose tolerance tests [14]. Obesity, insulin resistance, and T2DM are leading contributors to nonalcoholic fatty liver disease (NAFLD). Researchers are becoming increasingly aware that for several drugs, patients with NAFLD have drug-induced liver injury risk factors [15]. Regarding recent research, the incidence of liver disease was 4.42 times higher in patients with schizophrenia than in those without in a case-control study [16]. Furthermore, in schizophrenia patients, a previously executed study determined the chronic liver disease prevalence and incidence to be 1.27 and 1.15 times higher, respectively, when compared with that of the general population [17]. In primary rat hepatocytes and human hepatocytes, phospholipid, triglyceride, and free fatty acid synthesis were elevated through SGA administration [18,19]. In another recent study of SGAs, severe liver damage was sustained by rats given an antipsychotic drug; this damage manifested as fatty changes in the liver, vacuolar degeneration of hepatocytes, and a loss of usual architecture [20]. By contrast, mice treated with clozapine over 30 consecutive days had lower body weight and fasting glucose levels than had mice without clozapine treatment [21]. However, another study discovered that approximately two-thirds of patients taking clozapine did not develop diabetes mellitus, and that clozapine-treated patients had a risk profile linked to possible metabolic side effects and weight gain [22].

Individuals with diabetes mellitus are more likely to develop chronic kidney disease (CKD) [23,24]. SGAs are associated with increased risks of kidney injury and additional negative outcomes, including acute urinary retention and rhabdomyolysis or neuroleptic malignant syndrome [25]. These problems may be due to the effects of SGAs being associated with increased oxidative stress originating from damaged mitochondria [26,27]. Conversely, several case reports detailed cases of kidney failure attributed to interstitial nephritis in patients with chronic schizophrenia following the initiation of clozapine treatment, but the mechanism that leads to clozapine-related injury is not well understood [28,29]. The principal causes of dysfunction of the blood–retinal barrier and diabetes-induced retinal damage are prolonged hyperglycemia and insulin resistance [30,31]. In patients with schizophrenia and visual disturbance, deterioration of the visual cortex and retinal thickness was discovered within three years of the patients beginning to take an antipsychotic agent [32]. Other scholars reported that cystoid macular edema induced by risperidone may be caused by, for example, the agent directly affecting the retinal vascular endothelium or vasorelaxation through alpha-adrenergic blockade [33]. Thus, whether clozapine affects the retina should be investigated, and the possible mechanism should be identified.

The findings detailed in this Introduction indicate that clozapine exerts adverse effects probably by causing the development of metabolic syndromes, which contribute to body weight gain and hyperglycemia caused by clozapine. Accordingly, in the present study, we obtained a mouse obesity model by feeding the mice a high-fat diet (HFD) and administered clozapine to investigate what effects clozapine has in patients with obesity and psychotic disorders. Many reports discussed the side effects of clozapine on metabolism; in our study, we not only observed the glucose level, but also assessed the lipid metabolism and oxidative stress of the mice, and then explored the effects of an abnormal metabolism on the kidneys and eyeballs, particularly focusing on chromium. Chromium participates in the metabolism of proteins, lipids, and normal carbohydrates and has positive effects in individuals with renalpathy, obesity, glucose intolerance, or diabetes mellitus [34]. Our study also investigated how clozapine affects the hepatic and renal function of obese animals. The findings offer critical insight into how clozapine exerts metabolic effects and damages the kidneys, liver, and eyeballs when it is used as an antipsychotic drug over a long period; the findings also indicate whether clozapine exacerbates the harm induced to the liver, the kidneys, and the eyes by metabolic abnormalities, chromium changes, and oxidative stress.

## 2. Results

### 2.1. Clozapine Affects Food Efficiency, Food Intake, and Morphometric Parameters

We preliminarily examined mice fed a standard diet (SD); as revealed by the preliminary research results, clozapine treatment had nonsignificant antiobesity effects (as illustrated in Appendix A). The main experiment involved two groups, both of which were fed an HFD: the clozapine and control groups. After eight weeks of clozapine or saline administration, the clozapine group had altered morphometric parameters and higher serum leptin levels than did the control group (as illustrated in Figure 1). According to the growth curve of body weight, body weight was significantly greater in the clozapine group compared with that of the control group (as illustrated in Figure 1a). The average body weight of the clozapine group was 1.2 times that of the control group over the 8-week experimental period. The clozapine group had 2.2-fold higher weekly body weight gain than did the control group (as illustrated in Figure 1b). Additionally, the clozapine group exhibited significantly increased (by 20%) weekly food intake, as also occurred in the control group (as illustrated in Figure 1c). The daily food efficiency also increased significantly, but that of the clozapine group was determined to be superior to that of the control group by 1.8-fold (as illustrated in Figure 1d). An elevated concentration of leptin, which is a nutritional signal that regulates appetite, is associated with decreased appetite [35]. We discovered that the clozapine group demonstrated a 1.5-fold higher serum leptin level, on average, relative to the control group (as illustrated in Figure 1e), which is a contrasting finding to the food intake result.

### 2.2. Clozapine Increases Organ and Fat Pad Mass

The intergroup differences in body weight could were related to a change in body composition or adiposity, and this aspect was evaluated. After the 8-week experimental period, the clozapine group had significantly different body compositions to the control mice in terms of their retroperitoneal white adipose tissue (RWAT), kidney, epididymal white adipose tissue (EWAT), and liver masses but not in terms of the mass of their heart or spleen (as illustrated in Figure 2). After body weight normalization, the clozapine group was discovered to have significantly higher RWAT, EWAT, liver, and kidney masses (1.6-, 1.3-, 1.2-, and 1.2-fold higher, respectively) than did the control group. The ratios of kidney and heart mass to total body weight were not different, however.

### 2.3. Clozapine Increases Liver Fat Accumulation and Adipocyte Size

Tissues from mice in the clozapine and control groups were subjected to a staining procedure executed using hematoxylin and eosin (H&E), and the morphometry of the tissues was analyzed (as illustrated in Figure 3). Comparing the tissues obtained from the two groups revealed that the liver fat levels of the clozapine group tissues were markedly higher, and RWAT and EWAT adipocytes were larger in the clozapine group tissues (as illustrated in Figure 3a); these findings suggest that clozapine increased the hypertrophy of fat pads by blocking hepatic fat accumulation. The fatty liver scores of the clozapine group were almost 1.4 times those of the control group (*p* < 0.001; as illustrated in Figure 3b). The clozapine group had significantly bigger (1.2 and 1.3-fold bigger, respectively) RWAT and EWAT adipocytes; this finding agrees with the higher fat pad mass for the clozapine group. Regarding the number of RWAT and EWAT adipocytes in the tissues derived from the two groups, the tissues from the clozapine group had fewer adipocytes measuring 0–40 and 40–80 μm, but more adipocytes measuring 80–120 and >120 μm (as illustrated in Table 1). RWAT and EWAT adipocytes increase in size in animals fed an HFD, and clozapine was discovered to strengthen this effect.

### 2.4. Clozapine Increases ALT and AST Levels as Well as FABP4 mRNA and SREBP1 mRNA

Clozapine influenced hepatic function, including serum aspartate aminotransferase (AST) and alanine aminotransferase (ALT) levels. The ALT and AST levels of the clozapine group were 1.2 and 1.4 times those of the control group, respectively (as illustrated in Figure 4a,b). In humans and rats, fatty liver disease is promoted by sterol regulatory element–binding protein 1 (*SREBP1*) and fatty acid-binding protein 4 (FABP4) [34,36]. Significantly higher *FABP4* and *SREBP1* mRNA levels (2.2- and 3.2-fold higher, respectively; Figure 4c,d) were discovered in our clozapine group. The significant increases in AST and ALT levels were engendered by the genetic activation of hepatic steatosis; this may be because of the mediating effect of increased mRNA expression of fat-accumulation-related molecular mechanisms in the liver.

### 2.5. Clozapine Increases Serum and Hepatic Triglycerides, PNPLA3, and FASN But Decreases Hepatic Adiponectin

The serum and hepatic triglyceride levels (as illustrated in Figure 5a,b, respectively) were noted to be 1.7 and 2 times higher in the clozapine group compared with that of the control group. From Western blotting (as illustrated in Figure 5c), expression of fatty acid synthase (FASN) and patatin-like phospholipid domain containing protein 3 (PNPLA3) was discovered to be 2.0- and 1.3-fold higher in the clozapine group (as illustrated in Figure 5e,f, respectively). The liver enzyme FASN is crucial for fatty acid synthase and liver triglyceride metabolism [37]. In several liver diseases, an independent association exists between PNPLA3 and steatosis and fibrosis accumulation; additionally, PNPLA3 is associated with an elevated risk of hepatocellular carcinoma development in individuals with cirrhosis that developed from fatty liver disease [38]. Western blotting also revealed that the hepatic adiponectin level was 58.8% lower in the clozapine group (as illustrated in Figure 5d). Therefore, clozapine upregulated hepatic lipogenesis and increased the fatty liver scores, which were already high due to the HFD.

### 2.6. Clozapine Exacerbates Glucose Intolerance and Lowers Insulin Levels

Comparing the two groups demonstrated that glucose tolerance was impaired in the clozapine group (as illustrated in Figure 6a), supporting the finding that the clozapine group gained more weight and had higher fatty liver scores, both of which are known to usually result in lower insulin sensitivity (IS) [39]. Throughout the 120 min after glucose injection, the clozapine group had significantly increased levels of fasting blood glucose. At 120 min postinjection, the clozapine and control groups had glucose levels that were 18% and 16% higher than the baseline level, respectively, and the area under the curve (AUC) of glucose level in the clozapine group was 1.2 times that in the control group (as illustrated in Figure 6b). We discovered a large percentage of the clozapine group to be intolerant to glucose, defined herein as a blood glucose level—measured at 120 min postinjection—of >9 mmol/L (as illustrated in Figure 6c). Additionally, comparing the two groups indicated that the serum insulin levels of the clozapine group were significantly lower (as illustrated in Figure 6d). Accordingly, in our obese mice, clozapine resulted in higher severity of diabetes symptoms, such as hyperglycemia and impairment of hypoinsulinemia-associated glucose tolerance.

### 2.7. Clozapine Reduces IS by Attenuating Expression of Phosphorylated Akt and Glucose Transporter 4

The homeostatic model assessment for insulin resistance (HOMA-IR) and IS indices were employed to evaluate insulin resistance (IR) and IS, respectively [40], which were discovered to be 127% higher and 30% lower, respectively, in the clozapine group (as illustrated in Figure 7a,b). How clozapine affected insulin signaling in muscles was investigated (as illustrated in Figure 7c), and the results revealed that clozapine treatment engendered a reduction in Akt activation and glucose transporter 4 (GLUT4) expression (23% and 35% lower in the clozapine group, respectively; as illustrated in Figure 7d,e).

### 2.8. Effects on Organ and Tissue Chromium Levels and Urinary Chromium Loss

In glucose homeostasis, chromium is vital because it is involved in insulin activity and reduces IR [34,35]. Accordingly, in this study, whether the chromium levels of the clozapine group differed from those of the control group was investigated by determining the amount of chromium in various tissues harvested from the obese mice (as illustrated in Table 2). The clozapine group was discovered to have 1.2 times higher chromium intake, which supported the observation of hyperphagia in this group. In the clozapine group, the blood, bone, liver, muscle, and fat pads had 43%, 51%, 24%, 20%, and 28% lower chromium content at the end of the experiment, respectively, but the kidneys and urine had 1.7- and 2.0-times higher chromium content.

### 2.9. Clozapine Induces Renal Injury, Reduces the Amount of Antioxidant Enzymes in the Kidneys, and Raises Serum Creatinine and Blood Urea Nitrogen Levels

Because renal injury is likely to develop in individuals with diabetes, hyperlipidemia, and obesity, we investigated whether clozapine engenders kidney damage [41]. H&E staining revealed that clozapine induced glomerulonephritis that involved inflammatory cell infiltration (as illustrated in Figure 8a). Additionally, the clozapine group had significantly higher serum levels of creatinine and blood urea nitrogen (BUN; 2.3- and 2.0-fold higher, respectively; Figure 8b,c). The development of renal damage is related to the kidneys containing fewer antioxidant enzymes; such deceased antioxidant activity may cause necrosis and impaired renal function [42]. Compared with that of the control group, the clozapine group exhibited 25%, 33%, and 23% smaller amounts of the antioxidant biomarkers catalase (as illustrated in Figure 8d), glutathione peroxidase (GPx; as illustrated in Figure 8e), and superoxide dismutase (SOD; as illustrated in Figure 8f), respectively. Moreover, the kidneys of the clozapine group contained 1.6 times more reactive oxygen species (ROS; as illustrated in Figure 8g).

### 2.10. Clozapine Induces Retina Damage and Increases iNOS (Inducible Nitric Oxide Synthase), NF-κB (Nuclear Factor κB), and GLUT1 Expression But Reduces IκBα Expression in the Eyeballs

Diabetes-related complications such as retinal injury are postulated to involve elevation of oxidative stress, with such stress being a feature of diabetes [43,44]. H&E retinal section staining revealed that the inner plexiform and nuclear layers (IPL and INL, respectively) of the clozapine group were thinner than those of the control group; additionally, the ganglion cell layer was almost absent in the clozapine group (as illustrated in Figure 9a). Eyesight is frequently lost due to inflammation of the eye and associated complications. Evidence is mounting pertaining to the involvement of inflammation in the pathogenesis of various retinal diseases, such as diabetic retinopathy [45]. In our study, comparing the two groups demonstrated that *iNOS* and *NF-κB* expression was 6.2- and 2.8-fold higher, respectively, in the clozapine group (as illustrated in Figure 9b,c, respectively). The gene expression of *IκBα* was 80% lower in the clozapine group, however (as illustrated in Figure 9d). In addition, we investigated the gene expression of *GLUT1*, the primary facilitative glucose transporter for the retina [46]. *GLUT1* expression was 5.6-fold higher in the clozapine group (as illustrated in Figure 9e). These results indicate that clozapine elevated the risk of diabetic retinopathy by increasing the amount of inflammation and glucose uptake.

## 3. Discussion

Herein, we fed C57BL/6J mice an HFD diet, after which we probed the effect of clozapine on the consequent obesity development. Comparing the treatment and control groups revealed that the clozapine group was more obese, developed IR and glucose intolerance, acquired more visceral fat, and developed fatty liver disease. The clozapine group also had higher daily food efficiency; larger adipocytes; higher fatty liver scores and serum and hepatic triglyceride levels; and greater liver, kidney, and fat pad masses. This study thus confirmed that as well as increasing energy consumption and leading to greater weight gain, clozapine induces hypoinsulinemia, resulting in glucose homeostasis impairment. Higher fatty liver scores were discovered in the clozapine group, and scholars reported a link between this SGA and liver-based adipogenesis caused by related proteins (e.g., FASN and PNPLA3) as well as by the activation of SREBP1 and FABP4 in clozapine-treated mice. Lower activity of insulin signaling proteins is typically associated with changes in IR, and our clozapine group mice, which had obesity and hyperglycemia, exhibited impaired glucose homeostasis that was induced by reduced Akt phosphorylation and GLUT4 expression. In the clozapine group, the kidney function index was significantly higher and the amount of antioxidant enzymes significantly lower than those in the control group, corresponding to eventual renal injury. The clozapine treatment also resulted in retinal abnormality and increased the expression of inflammation-related genes, including *iNOS* and *NF-κB*, and glucose transporter *GLUT1* in eyeballs.

Weight gain is related to increased fat cell differentiation or increased fat pad weight caused by fat cell hypertrophy [35,47]. This was the reason that the clozapine group’s average body weight exceeded that of the control group; it also explains the relatively high masses of organs including the kidneys and liver, as well as the relatively high RWAT and EWAT masses in the clozapine group. Moreover, we found evidence that the body fat gain in the clozapine group was possibly caused by a decrease in energy expenditure due to uncoupling protein 1 (UCP1) being expressed within brown adipose tissue (as illustrated in Appendix A); energy expenditure is enhanced by UCP1 because this protein regulates the metabolic rate [48]. Comparing the two groups also revealed that the serum serotonin level was higher in the clozapine group; in a previous study, the glucose burning of brown fat was discovered to be inhibited when an excessive amount of peripheral serotonin was present in the blood, with this inhibition resulting in the development of diabetes and obesity [49] (as illustrated in Appendix A). Additionally, leptin action and IS partially mediate the relationship of obesity with excess energy intake [50,51]. Thus, clozapine treatment resulted in the food intake and leptin level of the clozapine group being higher than those of the control group (also fed an HFD). Additionally, the clozapine group had lower leptin receptor levels (as illustrated in Appendix A). Hence, long-term use of clozapine may cause the body to become leptin resistant and cause leptin signaling inhibition; this finding of the present study is valuable. Furthermore, the result regarding the daily food efficiency of the clozapine group is consistent with the findings regarding body weight and white adipose tissue weight.

Fatty liver disease is a consequence of hepatic fat accumulation. NAFLD, ranging from steatosis (with or without inflammation) and hepatocyte necrosis to cirrhosis, is a spectrum of diseases [52,53]. An elevated AST or ALT level is predictive of the presence of NAFLD [54]. Comparing the two groups in the present study revealed that the clozapine group of the present study exhibited higher AST and ALT levels and fatty liver scores. The expression of genes that engender de novo hepatic lipogenesis, lipid storage, and NAFLD pathogenesis was associated *SREBP1* and *FABP4* mRNA expression [55,56]. Moreover, one study discovered correlations of IR and liver inflammation and fibrosis severity with elevated FABP4 levels in patients with NAFLD [57]. As well as participating in the synthesis of hormones, SREBPs are involved in glucose and lipid homeostasis. In a mouse model of HFD-induced obesity, serum lipid levels and IS were improved through inhibition of SREBP expression [58]. Additionally, we analyzed the mRNA levels of pregnane X receptor (PXR) and ATP-binding cassette transporter isoform C2 (ABCC2), which play important roles in hepatic triglyceride accumulation and are critical mediators of liver steatosis and steatohepatitis development [59,60] (as illustrated in Appendix A). In our study, after clozapine treatment, *FABP4*, *SREBP1*, *PXR*, and *ABCC2* mRNA expression in the liver was higher, with the substantial lipid accumulation subsequently damaging the liver. Serum AST and ALT levels were subsequently evaluated.

Obesity is strongly associated with elevated triglyceride levels [61]. The factor most crucially underlying the development of NAFLD is lipid accumulation inside the liver, and augmented levels of triglyceride in circulation may be involved in limiting such accumulation [47,62]. We assessed the triglyceride content of serum and the liver. Our comparison of the two groups indicated that the clozapine group had augmented triglyceride levels in serum and the liver, which may reflect the clozapine-induced reduction of fibroblast growth factor 21 (FGF-21; a physiological energy sensor) activity (as illustrated in Appendix A). FGF-21 is involved in metabolism, angiogenesis, neural development, and cell proliferation through the exchange protein directly activated by cyclic adenosine monophosphate/protein kinase A pathway and in differentiation through the peroxisome proliferator-activated receptor (PPAR)-α pathway [63]. Long-term recombinant FGF-21 therapy lowers hepatic and serum triglyceride levels and decreases the fattiness of the liver in mice with diet-induced obesity; these actions may occur through suppression of *SREBP1*, a lipogenic gene [64]. Thus, our results indicate that a decreased FGF-21 level cannot suppress *SREBP1* expression or hepatic and serum triglyceride levels, consequently engendering obesity and fatty liver disease. Moreover, body weight regulation and obesity development may be affected by FASN [65]. One study revealed a significant correlation of FASN expression with the degree of steatosis in primary human hepatocytes in vitro as well as in experimental murine models and the livers of patients with NAFLD in vivo [66]. As expected, adiponectin was reduced in our clozapine-treated mice. Taken together, the findings indicate that clozapine treatment causes obesity and fat accumulation by creating metabolic abnormalities.

Clozapine has unique efficacy and is less likely to cause extrapyramidal side effects relative to typical antipsychotics; additionally, it is the only antipsychotic officially approved for schizophrenia that is resistant to therapy [67]. However, in oral glucose tolerance testing, clozapine is associated with significantly higher glucose levels when compared with typical antipsychotics [68]. Our IPGTT findings and calculations of AUCs for glucose plasma level in a T2DM and obesity mouse model reveal a clozapine-induced elevation of fasting glucose levels as well as a clozapine-induced dramatic worsening of glucose tolerance. Our results indicate that clozapine exacerbated T2DM by reducing the insulin level in serum; the 8-week clozapine treatment administered in this study resulted in lowered β-cell content in mouse pancreas sections (as illustrated in Appendix A). The signs of diabetes observed in this study when considering a long period of clozapine use suggest a possible association between glucose metabolism during obesity development and clozapine-induced glucose intolerance. Thus, in patients with T2DM and at high risk of psychotic disorders, the baseline blood glucose level should be monitored if the patient is undergoing clozapine treatment; medical practitioners should particularly be alert to the possibility of temporal hypoglycemia because it can result in difficulty recognizing diabetes.

Occurring in numerous tissues (e.g., the liver, muscle, the pancreatic islet, and adipose tissue) in individuals with obesity and involving considerable accumulation and inflammatory polarization of immune cells, inflammation was recognized as contributing to a dysfunctional metabolism, ultimately resulting in IR and T2DM [14,69]. Indeed, our comparison of the two groups revealed that the clozapine group exhibited higher serum levels of cytokines such as IL-1β and tumor necrosis factor-α (TNF-α; as illustrated in Appendix A). The progression of IR related to obesity can be accelerated by cytokine dysregulation, which also subsequently undermines IS and glucose homeostasis. The clozapine group’s lower insulin levels explain why the group’s IS index was relatively low. According to our findings, insulin signaling and glucose homeostasis may be impaired by clozapine through its lowering of GLUT4 expression and muscle Akt phosphorylation; in obese mice, metabolism is regulated by clozapine through its hepatic alteration of glucose flux and improvement of IS [70]. Notably, in adipose and muscle tissue, GLUT4 expedites glucose uptake stimulated by insulin, and relatively low expression of GLUT4 correlates with less extensive insulin-mediated glucose uptake [71]; exacerbation of hyperglycemia could be the reason for this phenomenon, as indicated by the relatively high glucose intolerance observed in the executed IPGTTs. Additionally, the pancreases of the clozapine group of the present study had lower SOD, GPx, and catalase activities than those of the control group (as illustrated in Appendix A). Indeed, antioxidant enzymes in pancreatic cells play critical roles in mediating IR, insulin secretion, and the late-stage complications of diabetes [72]. According to our data, clozapine aids development of IR by decreasing the activity of pancreatic antioxidant enzymes. Oxidative stress driven by a hyperglycemic state is often due to the generation of oxidative stress in various tissues and is a potential IR risk factor, which is indicated by results of impaired glucose tolerance [73]. In the long term, an HFD induces hyperglycemia; accordingly, according to the aforementioned findings, clozapine’s effect on IS could be due to the exacerbation of such hyperglycemia. This is because metabolism issues are worsened when insulin signaling is decreased, and these issues cause an individual to be unable to maintain glucose homeostasis.

In patients with T2DM, significant negative correlations were discovered between chromium level and hemoglobin A₁c and fasting glucose levels [71]. Chromium is physiologically stored in bone and then released and distributed as required to glucose-regulating tissues and organs, including adipose tissues, skeletal muscle, and the liver; this distribution ensures that blood glucose is controlled homeostatically by facilitating activation of insulin signaling [34,40]. Additionally, chromium movement participates in metabolism, specifically altering metabolism of glucose. Our findings indicate that relative to the control group mice, the clozapine group mice had significantly lower chromium in their muscles, livers, bones, epididymal fat pads, and blood despite higher chromium intake being linked to hyperphagia in those taking clozapine. Studies also showed that the administration of SGA negatively affects chromium accumulation in selected metabolic organs and tissues [34,35]. Additionally, a low serum chromium level was discovered to elevate the risk of T2DM-related complications and greatly contribute to higher glycemic status [74]. In our obese mice, clozapine possibly dictated how chromium was distributed among tissues. Moreover, the livers and muscle—the location of most glucose metabolism—of the clozapine group contained less chromium than did those of the control group [28,42]; hence, the essential metal chromium did not negate the hyperglycemia occurring upon clozapine treatment. Thus, in different tissues and organs, chromium levels are clearly variously affected by long-term clozapine use. Nevertheless, our findings regarding chromium mobilization and redistribution revealed that chromium cannot easily mitigate hyperglycemia that is exacerbated by clozapine treatment.

The major percentage of trace elements is absorbed at the proximal renal tubule, but a certain amount is excreted in urine [35]. The kidneys and urine of our clozapine group had high chromium content, whereas the content in the blood and metabolic tissues was low. Thus, clozapine resulted in more chromium being lost through urine after its movement to the kidneys from metabolic tissues. Impaired renal function and kidney injury responses are generally indicated by more trace metals being excreted and less trace metals being reabsorbed [34,75]. Our clozapine group had higher kidney function indices (i.e., creatinine and BUN levels) than did the control group, supporting this phenomenon. Additionally, injury lesions, along with glomerulonephritis, were discovered to form in the kidneys of our clozapine group. The aforementioned renal injuries were probably related to the negative total chromium balance and greater chromium loss through urine in our clozapine group. The glucose intolerance of the mice was exacerbated by the substantial chromium deficit. Thus, in patients with T2DM and schizophrenia, researchers must discover whether these effects occur when other SGAs—provided along with clozapine or used as a replacement for clozapine for the reduction of renal adverse effects—are employed.

In the present study, glomerulonephritis was discovered in the clozapine group; similar findings were obtained in other studies, with several psychotropic drugs (e.g., ziprasidone, olanzapine, chlorpromazine, and aripiprazole) found to injure the kidneys and cause their dysfunction (through, for example, CKD, interstitial nephritis, and glomerulonephritis) [76,77]. Renal function is commonly evaluated using serum creatinine and BUN levels [78], and in patients with moderate–severe CKD, disease progression was associated with higher creatinine and BUN levels [79]. Kidney damage can thus be concluded as being caused by long-term clozapine use; this damage is accompanied by inflammatory cells infiltrating the renal interstitium and by elevated levels of major nephropathy indicators, such as BUN and creatinine. To ensure that such kidney damage is identified in a timely manner, clinicians should closely monitor the renal function of patients receiving clozapine. Even at the early stage of CKD, an association exists between a graded increase in oxidative stress markers and kidney disease severity, and renal disease can result from more numerous ROS and weaker antioxidant defense [80]. Therefore, antioxidant enzyme activity can benefit the kidneys. Our clozapine group of HFD-fed mice had lower SOD, GPx, and catalase activities than our control mice. In addition, the *iNOS* mRNA level (as illustrated in Appendix A) and immunohistochemical (IHC) staining showed that the levels of renal inflammatory cytokines, such as IL-1β (as illustrated in Appendix A), were high in the clozapine group. The data suggest that renal injury due to hyperglycemia can be caused by continual clozapine administration over a long period, as can chronic inflammation; this injury and inflammation worsen the injury induced by oxidative stress and lowered antioxidant levels. Therefore, in psychotic disorders, antipsychotropic medications appear to be associated with the side-effect hyperglycemia and a certain kidney damage profile rather than altered antioxidant parameters.

In diabetes, the retina is under high oxidative stress, which scholars concluded is involved in retinopathy development [81]. A substantial chronic complication of T2DM, diabetic retinopathy ultimately results in blindness [82]. Diabetes-associated retinal damage and dysfunction of the blood–retinal barrier are mostly caused by IS and prolonged hyperglycemia [30,83]. In humans and a rodent model of diabetes, the proinflammatory protein iNOS was discovered to be upregulated in the retina [84]. Additionally, the inducible transcription factor NF-κB, which is inhibited by IκBα, is widely expressed and crucially regulates several genes that participate in immune- and inflammation-related responses as well as cell proliferation and apoptosis; it is activated by proinflammatory proteins, including iNOS [85]. In a rodent model of diabetes, the disease was discovered to activate NF-κB in the retina [86]. No protein other than GLUT1 is known to transport glucose across the blood–retinal barrier, and in ganglion cells, GLUT1 is responsible for how glucose is distributed [46,87]. Furthermore, GLUT1 holds promise as a diabetic retinopathy target [88]. In this study, H&E staining revealed that the retinal cells of the clozapine group mice were more severely damaged when assessed against those of the control mice; in RT-PCR, the gene expression of inflammatory markers, including *iNOS* and *NF-κB*, and the glucose transporter *GLUT1* were lower in the clozapine-treated mice. In brief, clozapine treatment caused an increase in inflammation and of glucose uptake in the eyeball, resulting in more severe diabetes and retinopathy.

A mouse model of obesity and hyperglycemia was successfully created through an HFD. Clozapine treatment of the obese mice resulted in higher food intake; more weight gain; greater fat pad weight; higher fatty liver scores; larger adipocytes; and higher serum levels of AST, ALT, and triglyceride. These results all suggest lipid accumulation and the prompting of lipogenesis with the objective of nutrient metabolism regulation. The hyperglycemia of the clozapine group mice was exacerbated, and their glucose tolerance was reduced; they had interstitial nephritis as well as lowered expression of GLUT4 in skeletal muscles. Retinal *GLUT1* expression perhaps affected the aforementioned alteration of glucose metabolism. The clozapine group had poorer IS and worsened glucose homeostasis. This effect on glucose homeostasis was strengthened by the HFD the mice received, and diabetes symptoms developed more rapidly in the clozapine group. Our findings on clozapine indicate side effects related to renal damage, T2DM, retinopathy, and fatty liver disease and imply worsening of diabetic retinopathy, CKD, obesity, and diabetes when clozapine is used in the long term.

## 4. Materials and Methods

### 4.1. Animals, HFD-Induced Obesity, and Clozapine Treatment

From the Education Research Resource, National Laboratory Animal Center, Taipei, Taiwan, we obtained male 5-week-old C57BL/6J mice, which we housed in accordance with the Guidelines for the Care and Use of Laboratory Animals and the recommendations of the Taiwanese government; our executed experiments also accorded with these guidelines and recommendations. National Chiayi University’s Institutional Animal Care and Use Committee ratified our implemented experimental protocol (approval no. 109019 on 24 February 2020).

The HFD diet employed in this study was diet 592Z, comprising 20.4% protein and 35.5% lard (amount of metabolizable energy, 4.5 kcal/g; PMI Nutrition International, Brentwood, MO, USA). The mice were continually fed this HFD for 10 weeks, and the duration of the HFD feeding was designed to be 10 weeks longer than that typically employed (i.e., 4 weeks) to ensure that obesity developed [70]. After mice were fed an HFD for 10 weeks, the control group was administered saline through daily gavage, whereas the clozapine group received 2 mg/kg/day oral clozapine (Sigma–Aldrich, St. Louis, MO, USA) for 8 weeks while being fed the HFD. The final average body weight of these groups was 43.92 ± 1.57 and 53.72 ± 2.79 g, respectively (*p* < 0.01). When deciding on the dose of clozapine, we reviewed the related literature on cardiovascular disease, immunosuppression, behavior, and antipsychotic disease in mouse models [70,89]. Before commencing the study, we preliminarily administered an oral dose of 1 mg/kg; however, this resulted in body weight and weight gain that were not significantly different from those of the control group of obese mice (as illustrated in Appendix A). The clozapine dose 2 mg/kg was thus selected. The mice had ad libitum food and water access throughout the experiment. Microisolation cages were employed to separately house the animals. The cages were placed on racks subject to ventilation employing highly efficient particulate air filters. A 12–12-h light–dark cycle was applied. The temperature was maintained at 22 ± 1 °C, with the humidity being maintained at 55% ± 5%. Throughout the experiment, weekly monitoring of food intake and body weight was executed.

All mice were euthanized at the finish of the experiment. Various tissues and their blood were harvested.

### 4.2. Measurement of Food Intake, Body Weight, and Insulin and Leptin Levels

As mentioned, weekly monitoring of food intake and body was executed. We calculated the amount of food consumed by weighing the food left by the mice in their dispenser and on the floor. Using the harvested blood and tissues, serum insulin and leptin levels were measured using enzyme-linked immunosorbent assay kits (#90030 and #INSKR020; Crystal Chem, Downers Grove, IL, USA).

### 4.3. Measurement of Serum AST, ALT, BUN, Creatinine, and Triglyceride Levels and Hepatic Triglyceride Levels

A Catalyst One Chemistry Analyzer (IDEXX Laboratories, Westbrook, ME, USA) and commercial kits were employed, following the manufacturer instructions, to measure AST, ALT, BUN, triglyceride, and creatinine levels in blood samples. Inter- and intra-analyses had coefficients of variation < 2%. Triton X-100 solution was employed to obtain hepatic triglycerides from homogenized liver samples, as was done in another study [90]. Extract solubilization was then performed; the extracts were twice heated gradually to 90 °C over 5 min and subsequently cooled to room temperature. Insoluble material was removed through centrifugation. Lastly, we obtained the supernatant and employed the Triglyceride Quantification Kit from BioVision (Milpitas, CA, USA) to subject it to colorimetric assay-based triglyceride analysis.

### 4.4. IPGTT

The IPGTT was performed once the clozapine or saline was administered for 7 weeks. The mice were fasted overnight prior to the IPGTT, although they had free water access. The glucose dose in the IPGTT was 1 g of glucose/kg body weight. Blood was obtained from the mouse tail 0, 30, 60, 90, and 120 min postinjection, and a One Touch glucose meter (LifeScan, Milpitas, CA, USA) was used for blood glucose level measurement. The relevant AUC (0–120 min) was determined to assess how tolerant the two groups were to glucose.

### 4.5. IR and IS Indices

Estimation of the HOMA-IR index and IS index by using fasting glucose is common in the literature [35,70]. Mouse IR and insulin function were thus assessed using these two indices, which were calculated [70] as follows:HOMA-IR index = [fasting insulin (in mU/L) × fasting glucose (in mmol/L)]/22.5
IS index = 1000/[fasting insulin (in mU/L) × fasting glucose (in mmol/L)].

Fasting plasma insulin and glucose levels were derived using the clamp-measurement-validated HOMA method.

### 4.6. Tissue Histology and Morphometry

We weighed the RWAT, EWAT, spleen, kidneys, heart, and liver and express them herein as a percentage of overall body weight. H&E staining was executed to determine the degree of hepatic fat infiltration, which was categorized as follows: 0%, <5%, 5–25%, 25–50%, and >50% fat infiltration of the liver surface equal to scores of 0, 1, 2, 3, and 4, respectively [35,70]. We collected several sections of retroperitoneal and epididymal adipose tissues and discovered the size of adipocytes by systematically analyzing them. The sections were stained with H&E. For each sample, we examined at least 10 fields (~100 adipocytes) within each slide [35,70]. Additionally, the kidneys were bisected along the longitudinal axis to yield tissues, which were then also stained with H&E. In accordance with an established method [91], we concealed the specimens’ origin in the detection of interstitial nephritis. Images were obtained using the high-resolution digital microscope Moticam 2300 (Motic Instruments, Richmond, BC, Canada); we employed the accompanying software, Motic Images Plus (version 2.0), to obtain adipocyte size distributions. In addition, the right eyeball was fixed in 4% paraformaldehyde solution, followed by dehydration by passing it over a series of graded ethanol concentrations (70%, 80%, 95%, and 100%, sequentially). Dehydrated eyeballs were then treated with xylene to render them transparent. After paraffin embedding was applied, the eyeballs were cut into sections for H&E staining.

### 4.7. Extraction of RNA and Real-Time Quantitative Polymerase Chain Reaction

The total RNA in liver and eyeball tissues was extracted using TRI Reagent (Sigma–Aldrich). Absorbance, investigated using a Qubit fluorometer (Invitrogen, Carlsbad, CA, USA), was obtained at 260–280 and 230–260 nm for assessing the concentration of RNA. The iScript complementary DNA (cDNA) synthesis kit (Bio-Rad, Hercules, CA, USA) was applied, following the manufacturer’s instructions, to reverse transcribe 1 μg of RNA into cDNA. This cDNA and the iTaq universal SYBR Green Supermix (Bio-Rad) were then employed for real-time polymerase chain reaction (PCR). The CFX Connect Real-Time PCR System (Bio-Rad) was used to obtain the expression levels of *IκBα*, *NF-κB*, *GLUT1*, *iNOS*, *SREBP1*, and *FABP4* mRNA. The following details the PCR procedure: 95 °C for 5 min; 45 cycles at 95 °C for 15 s; and finally, 60 °C for 25 s. The following sequence primers were utilized: *FABP4*: forward, 5′-GATGAAATCACCGCAGACGACA-3′, and reverse, 5′-ATTGTGGTCGACTTTCCATCCC-3′ [92]; *SREBP1*: forward, 5′-CGG AAGCTGTCGGGGTAG-3′, and reverse, 5′-GTTGTTGATGAGCTGGAGCA-3′ [93]; *iNOS*: forward, 5′- CCTCCTCCACCCTACCAAGT-3′, and reverse, 5′- CACCCAAAGTGCTTCAGTCA-3′ [94]; *GLUT1*: forward, 5′-TCAACACGGCCTTCACTG-3′, and reverse, 5′-CACGATGCTCAG ATAGGACATC-3′ [95]; *NF-κB*: forward, 5′- GCAACTCTGTCCTGCACCTA -3′, and reverse, 5′-CTGCTCCTGAGCGTTGACTT-3′ [96]; *IκBα*: forward, 5′- AAGTGATCCGCCAGGTGAAG -3′, and reverse, 5′- CTGCTCACAGGCAAGGTGTA -3 [97]; *β-actin*: forward, 5′- GGCTGTATTCCCCTCCATCG -3′, and reverse, 5′- CCAGTTGGTAACAATGCCATGT -3 [98]. We computed the level of expression of the target gene relative to the *β-actin* level; expression levels are presented using the 2^−ΔΔCt^ method.

### 4.8. Western Blotting

An overdose of anesthetic in combination with CO_2_ was used for euthanization. We obtained the gastrocnemius and liver muscles rapidly and then coarsely minced and homogenized them. Western blotting was implemented using a reported method [35]. We purchased antibodies against Akt, phospho-Akt (Ser473), actin, GLUT4, adiponectin, and FASN from Cell Signaling Technology (Beverly, MA, USA), and antibodies against PNPLA3 came from Sigma–Aldrich. Immunoreactive signals were obtained through enhanced chemiluminescence reagents (Thermo Scientific, Rockford, MA, USA) and then detected through UVP ChemStudio (Analytik Jena, Upland, CA, USA). ImageJ, a program designed by the National Institutes of Health (Bethesda, MA, USA), was employed for quantifying protein expression and phosphorylation.

### 4.9. Chromium Concentration Analysis

Blood, urine, and various types of tissues (kidney, fat, muscle, liver, bone, and blood) were collected after the experiments. After saline rinsing, we blotted the tissue samples dry and weighed them. The method outlined in our prior article was employed for calculating the chromium concentration of the samples [40]. Each sample (0.1 g of tissues and 25 µL of urine and blood) was subject to overnight digestion with 1 mL of 65% nitric acid at 100 °C. Distiller water was then used to dilute the digested solution to a 5-mL solution in advance of measurements. Graphite furnace atomic absorption spectrophotometry (Z-2000 series polarized Zeeman model; Hitachi, Tokyo, Japan) was employed to obtain CR levels. We used analysis lines 359.3 for chromium, and we express concentrations in nanograms per gram (for tissues) or per milliliter (for fluids). The relative rate of chromium recovery was calculated at 5 ppb of the limits of the quantification levels by 4% (*n* = 5). The absorption data were plotted onto a 1–500-ppb standard curve, and regression analysis was performed to identify the total chromium level in samples at R^2^ > 0.995.

### 4.10. Renal ROA, SOD, GPx, and Catalase Level Measurements

Homogenized kidney lysates were prepared using 0.1 M Tris/HCl (pH 7.4) containing 0.5% Triton X-100, 5 mM β-mercaptoethanol, and 0.1 mg/mL phenylmethanesulfonylfluoride. Fresh renal cortical tissue was subsequently removed, as detailed elsewhere [99]. Centrifugation was performed at 14,000× *g* at 4 °C for 5 min, and the supernatant was obtained. A colorimetrical kit (BioVision, Milpitas, CA, USA) was then used, following the manufacturers’ instructions, to measure the SOD, GPx, and catalase content of kidney samples (kits #K335-100, #K762-100, and #K773-100, respectively). Regarding ROS level measurement, weighing, homogenization using phosphate-buffered saline on ice, and centrifugation (5000× *g*, 4 °C, 10 min) of the kidney tissues were conducted [78]. A colorimetrical kit (#KTE71621) from Abbkine (Redlands, CA, USA) was utilized to measure the ROS content of the supernatant obtained through centrifugation.

### 4.11. Statistical Analysis

We express all of our derived data herein as the mean ± standard deviation and employed the *t* test to identify intergroup differences. Additionally, we applied analysis of variance and then post hoc Bonferroni testing to determine differences among three or more groups. The level of statistical significance was 0.05. The significance in contingency data was assessed using the Fisher exact test.

## 5. Conclusions

The present study suggests that clozapine potentially has adverse side effects in obese mice. Our findings, obtained in an HFD-induced mouse model of obesity, reveal worsened obesity and hyperglycemia as well as kidney damage when clozapine is taken continually. Increased fatty liver scores and lower glucose transporter expression indicate that these side effects were related to less successful metabolic homeostasis. Our clozapine group differed from our control group in weight gain, food efficiency, adipocyte size, fat tissue weight, renal and retinal pathology, serum AST and ALT levels, serum and hepatic triglyceride levels, and fatty liver disease severity. A person may become more susceptible to experiencing obesity-related disturbance of their metabolism when their adipose lipogenesis regulation is worsened and hepatic lipid accumulation occurs. Our data suggest that clozapine reduced the amount of pancreatic antioxidant enzymes, increased inflammatory changes, and resulted in β-cell damage. Moreover, our derived findings confirm clozapine-induced aggravation of hyperglycemia and demonstrate that clozapine-engendered lowering of GLUT4 expression and Akt phosphorylation in insulin signaling was related to the augmentation of IR, impairment of glucose tolerance, and reduction of IS. Exacerbation of hyperglycemia was possibly the cause of the clozapine group’s nephropathy, which could may be worsened by the smaller amount of antioxidant enzymes and higher ROS levels in the kidney compared with the control group. The clozapine group had diabetic retinopathy and thinner INL and IPL due to higher inflammatory *iNOS* and *NF-κB* expression. Thus, clozapine’s adverse effects on obesity development, renal injury, retinal injury response, and hyperglycemia in our clozapine group support the notion of clozapine being an SGA that increases the risk of metabolic abnormalities. After treatment with clozapine is initiated, clinicians should monitor not only blood glucose, but also eyesight and liver and kidney functions to ensure that this potential side effect is recognized quickly, particularly in individuals with schizophrenia. Rapid recognition would mean that hyperglycemia, obesity, and cachexia could be prevented.

## Figures and Tables

**Figure 1 ijms-22-06680-f001:**
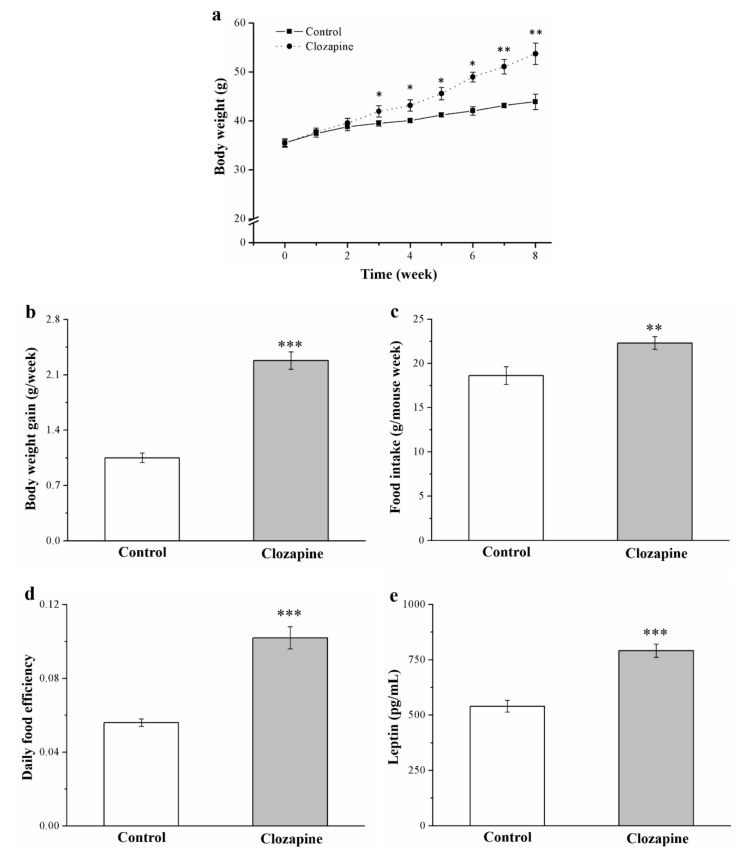
(**a**) Change in body weight, (**b**) weekly weight gain per mouse, (**c**) weekly food intake per mouse, (**d**) daily food efficiency, and (**e**) serum leptin level in clozapine group (receiving 2 mg/kg/day) and control group for 8-week experimental period. Data are mean ± standard deviation (*n* = 10). * *p* < 0.05, ** *p* < 0.01, *** *p* < 0.001.

**Figure 2 ijms-22-06680-f002:**
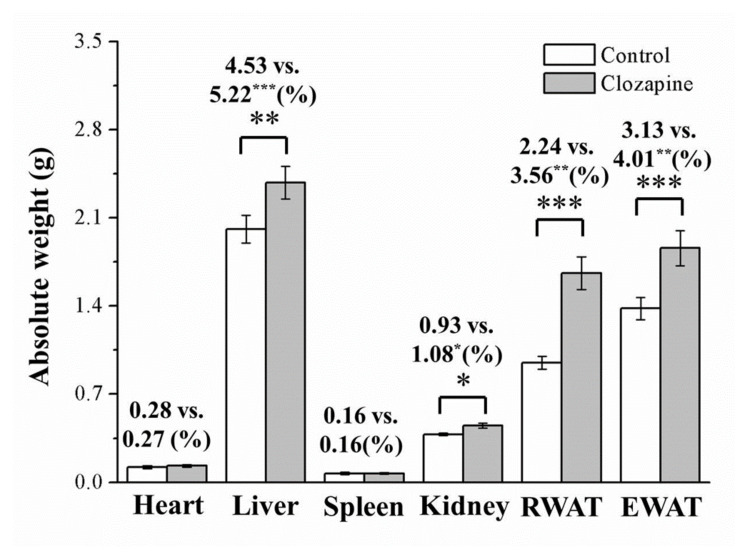
Mass of epididymal white adipose tissue (EWAT), retroperitoneal white adipose tissue (RWAT), and organs in clozapine (receiving 2 mg/kg/day) and control groups after 8-week experimental period. Numbers above bars indicate masses normalized by body weight. Data are mean ± standard deviation (*n* = 10). * *p* < 0.05, ** *p* < 0.01, *** *p* < 0.001.

**Figure 3 ijms-22-06680-f003:**
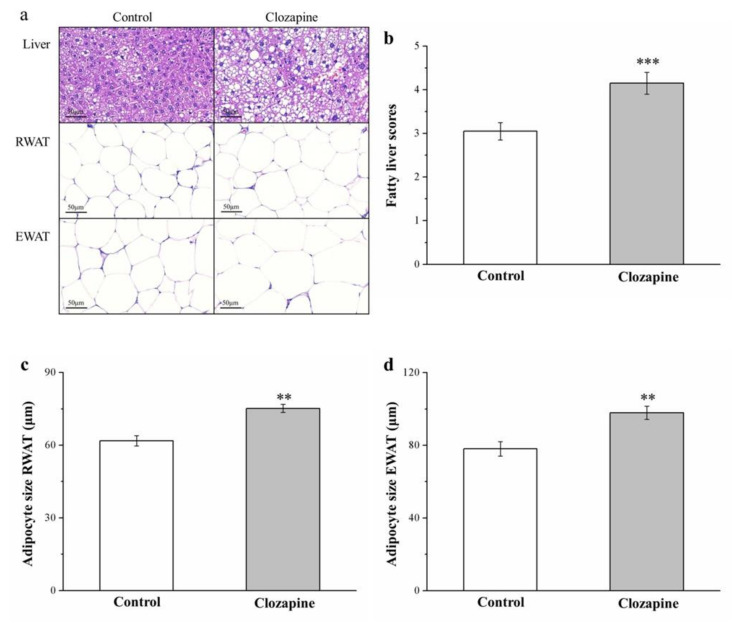
(**a**) Hematoxylin and eosin (H&E)-stained sections of liver, EWAT, and RWAT tissues (magnification, 200×); (**b**) fatty liver score; and size of (**c**) RWAT and (**d**) EWAT adipocytes for clozapine (receiving 2 mg/kg/day) and control groups. Data are mean ± standard deviation (*n* = 10). ** *p* < 0.01, *** *p* < 0.001.

**Figure 4 ijms-22-06680-f004:**
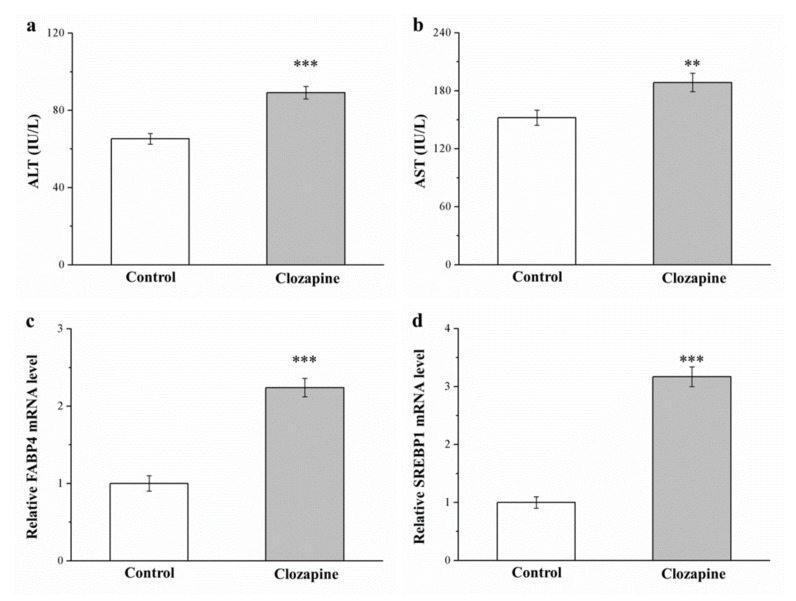
Serum (**a**) alanine aminotransferase (ALT) and (**b**) aspartate aminotransferase (AST) levels and (**c**) *FABP4* and (**d**) *SREBP1* mRNA levels in livers derived from clozapine and control groups after 8-week experimental period. Data are mean ± standard deviation (*n* = 10). ** *p* < 0.01, *** *p* < 0.001.

**Figure 5 ijms-22-06680-f005:**
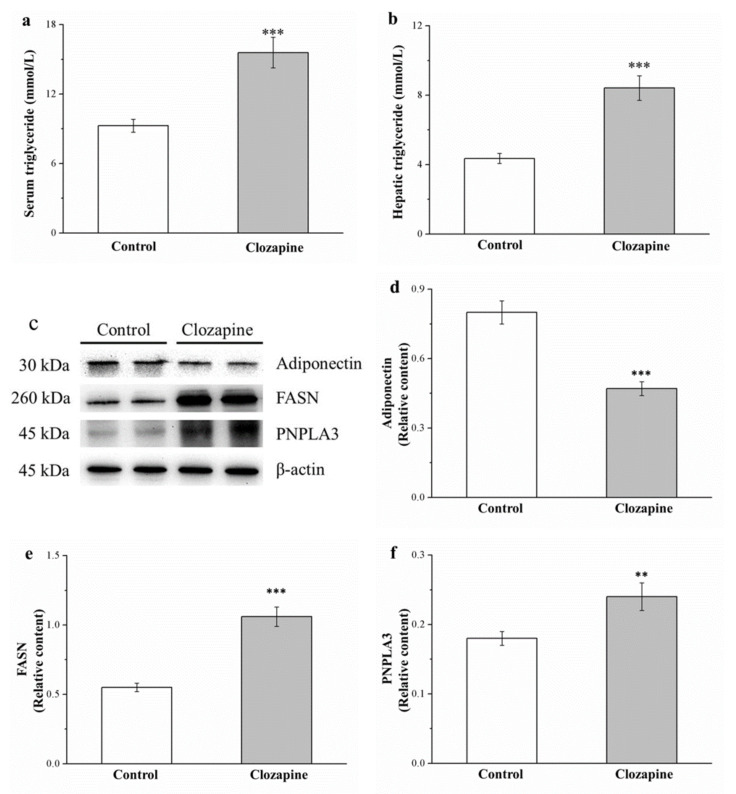
(**a**) Serum and (**b**) hepatic triglyceride levels. (**c**) Representative Western blot of liver extracts. (**d**) Adiponectin, (**e**) FASN, and (**f**) PNPLA3 expression. Data are mean ± standard deviation (*n* = 10). ** *p* < 0.01, *** *p* < 0.001.

**Figure 6 ijms-22-06680-f006:**
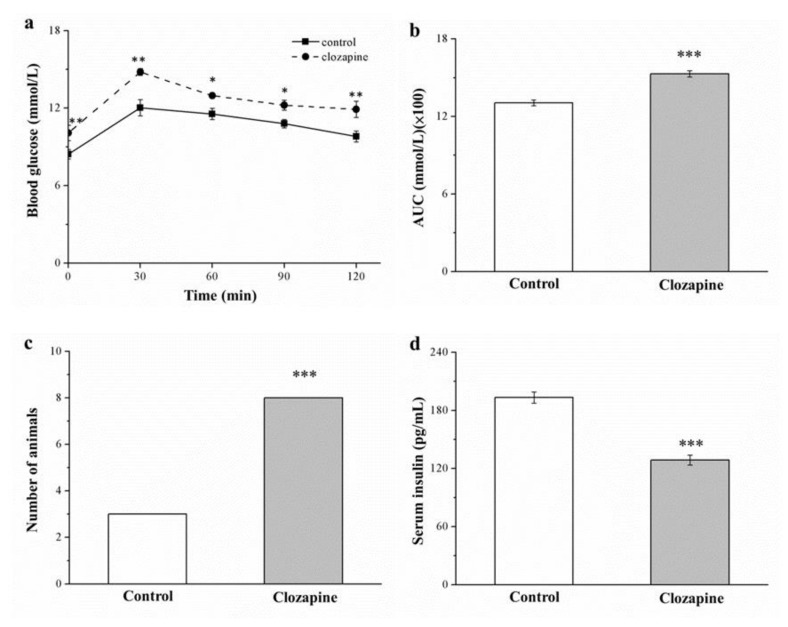
(**a**) Intraperitoneal tolerance to 1 g of glucose per kilogram of body weight; (**b**) area under curve at 120 min following injection of glucose; (**c**) number of mice with glucose intolerance (Fisher’s exact test); and (**d**) serum insulin levels in clozapine (receiving 2 mg/kg/day) and control groups. Data are mean ± standard deviation (*n* = 10). * *p* < 0.05, ** *p* < 0.01, *** *p* < 0.001.

**Figure 7 ijms-22-06680-f007:**
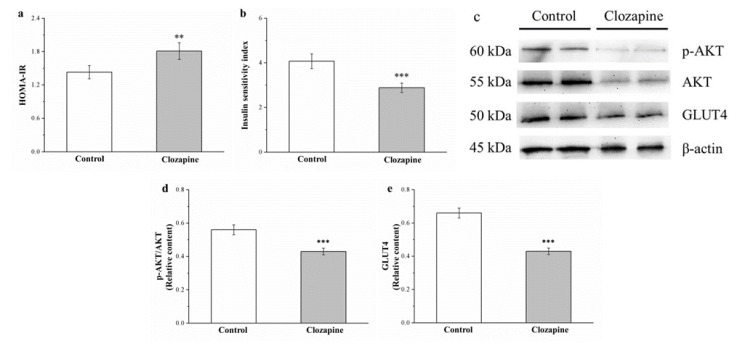
(**a**) Homeostatic model assessment for insulin resistance (HOMA-IR) and (**b**) insulin sensitivity indices; (**c**) representative blot of muscle extracts; and (**d**) Akt phosphorylation and (**e**) GLUT4 expression in gastrocnemius muscle in clozapine (receiving 2 mg/kg/day) and control groups for 8-week experimental period. Data are mean ± standard deviation (*n* = 10). ** *p* < 0.05, *** *p* < 0.001.

**Figure 8 ijms-22-06680-f008:**
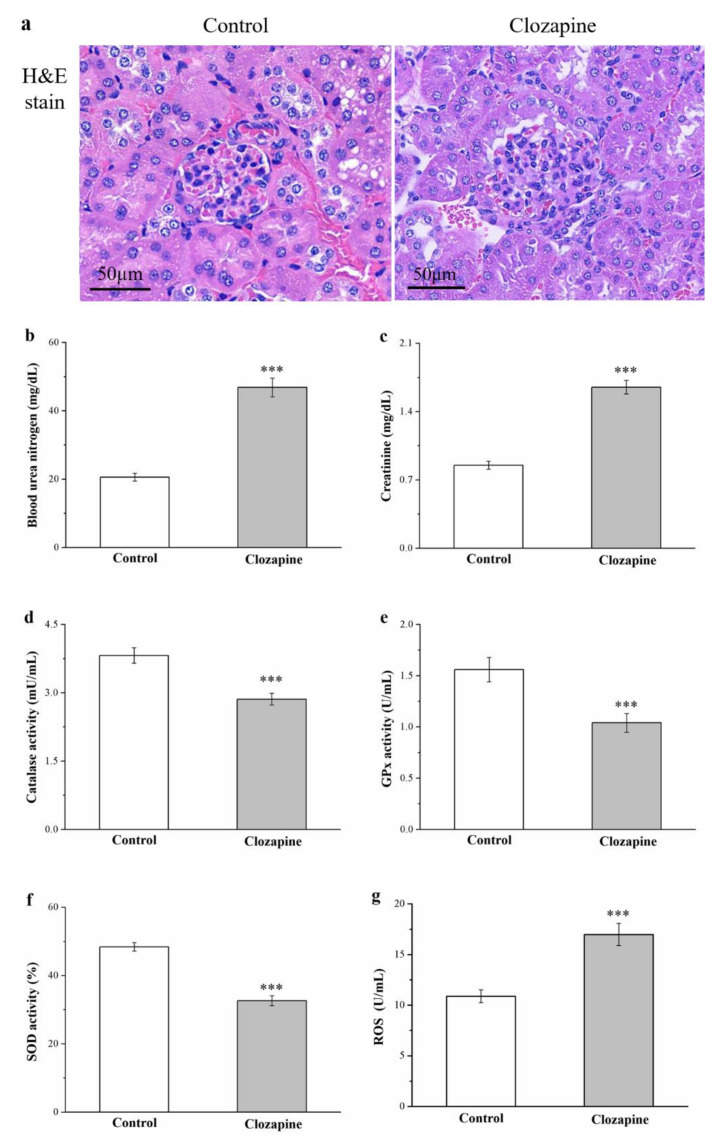
(**a**) Hematoxylin and eosin (H&E) stains showing differing renal morphology (magnification, 200×); serum levels of (**b**) blood urea nitrogen and (**c**) creatinine; (**d**) renal catalase activity; (**e**) renal glutathione peroxidase (GPx) activity; (**f**) renal superoxide dismutase (SOD) activity; and (**g**) renal reactive oxygen species (ROS) in the clozapine (receiving 2 mg/kg/day) and control groups. Data are mean ± standard deviation (*n* = 10). *** *p* < 0.001.

**Figure 9 ijms-22-06680-f009:**
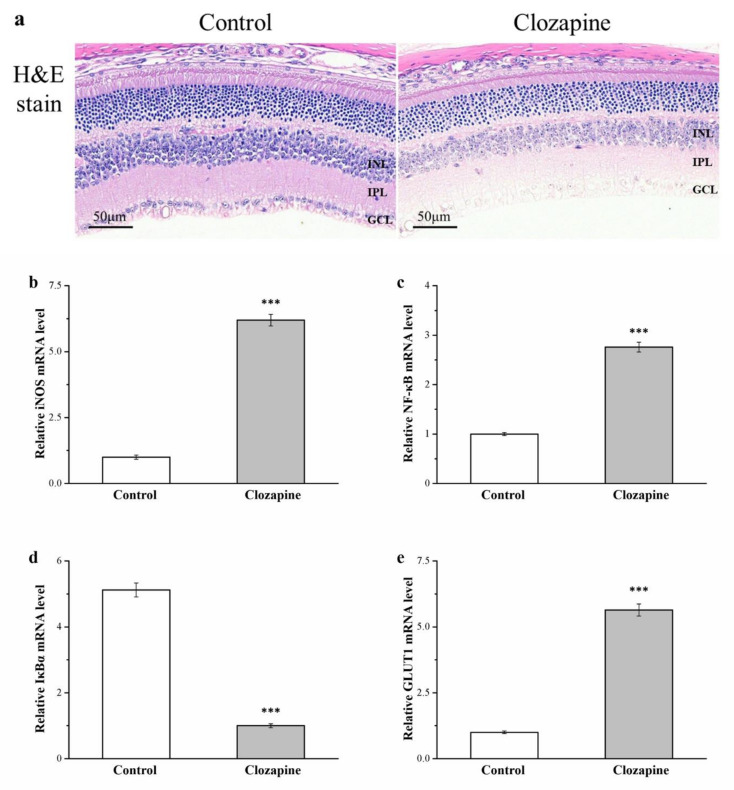
(**a**) Hematoxylin and eosin (H&E) staining of retina (magnification, 200×) and (**b**) *iNOS*, (**c**) *NF-κB*, (**d**) *IκBα*, and (**e**) *GLUT1* mRNA levels in eyeballs derived from clozapine and control groups. INL, inner nuclear layer; IPL, inner plexiform layer; GCL, ganglion cell layer. Data are mean ± standard deviation (*n* = 10). *** *p* < 0.001.

**Table 1 ijms-22-06680-t001:** Effects of clozapine on fat cell size distribution for clozapine (receiving 2 mg/kg/day) and control groups.

Variable	Control	Clozapine
RWAT		
Adipocyte diameter		
0–40 μm (%)	13.88 ± 0.65	0 ± 0 ***
40–80 μm (%)	50.01 ± 1.25	25.64 ± 1.07 ***
80–120 μm (%)	36.11 ± 0.42	48.71 ± 1.12 ***
>120 μm (%)	0 ± 0	25.64 ± 0.96 ***
EWAT		
Adipocyte diameter		
0–40 μm (%)	9.09 ± 0.47	0 ± 0 ***
40–80 μm (%)	77.27 ± 0.94	27.27 ± 0.57 ***
80–120 μm (%)	13.63 ± 1.03	38.18 ± 1.69 ***
>120 μm (%)	0 ± 0	56.67 ± 1.42 ***

Data are presented as mean ± standard deviation (*n* = 10). *** *p* < 0.001.

**Table 2 ijms-22-06680-t002:** Tissue and organ chromium levels in clozapine and control groups after 8-week experimental period.

Variable	Control	Clozapine
Chromium intake/mouse/week (μg)	20.88 ± 1.29	25.00 ± 0.80 **
Blood (ng/mL)	175.32 ± 8.51	100.24 ± 5.85 ***
Bone (ng/g)	382.37 ± 11.26	185.57 ± 7.62 ***
Liver (ng/g)	78.35 ± 6.27	59.48 ± 4.16 **
Muscle (ng/g)	53.21 ± 3.98	42.36 ± 3.47 **
Epididymal fat pads (ng/g)	49.63 ± 3.15	35.52 ± 2.71 ***
Kidney (ng/g)	100.49 ± 5.68	172.27 ± 4.15 ***
Urine (ng/mL)	48.26± 2.13	98.38 ± 2.48 ***

Data are mean ± standard deviation (*n* = 10). ** *p* < 0.01, *** *p* < 0.001.

## Data Availability

The data presented in this study are available on request from the corresponding author.

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
