# Peer review of "Clozapine Worsens Glucose Intolerance, Nonalcoholic Fatty Liver Disease, Kidney Damage, and Retinal Injury and Increases Renal Reactive Oxygen Species Production and Chromium Loss in Obese Mice"

_ijms, 2021, doi:10.3390/ijms22136680_

Round 1

Reviewer 1 Report

The research article “Clozapine Worsens Glucose Intolerance, Nonalcoholic Fatty Liver Disease, Kidney Damage, and Retinal Injury and Increases Renal Reactive Oxygen Species Production and Chromium Loss in Obese Mice” is dedicated to investigate how clozapine affects weight, the bodily distribution of chromium, liver damage, fatty liver scores, glucose homeostasis, renal impairment, and retinopathy in mice fed a high fat diet. The article is well written. The study has a good design. The article is logically divided into sections and subsections. In the article there are no grammatical and stylistic errors. There are many tables and figures of good quality presented in the article. The references cited are relevant and adequate. The work has a high degree of novelty. In my opinion, this review paper can be recommended for publication after minor revision. It is recommended to expand section “4.2. Measurement of food Intake, body weight, and insulin and leptin levels”. It is recommended to include a list of abbreviations, used in the article. It is recommended to add articles of 2021 to the list of references.

Reviewer 2 Report

The present study by Dr. Chang and co-authors investigates the role of clozapine in context of high fat diet (HFD) in mice. The data provide convincing evidence that using of clozapine significantly affect adipose tissue, liver function, retina, metabolic state and somehow kidney function in HFD-fed mice. The work mainly provides in vivo data. While this work does not provide a mechanistical support of a suggested mechanism of clozapine action, it still may present an interest for other researchers. However, there are some questions that need to be addressed:

  1. Not clear if the animals used in the study were fed on HFD for 10 weeks before clozapine treatment plus 8 weeks during clozapine treatment (18 weeks in total) or they were on HFD for 2 weeks plus 8 weeks during clozapine treatment (10 weeks in total). Please clarify. It may represent a particular interest as HFD duration affects development of diabetes-associated complications.
  2. It would be informative to provide glucose/body weight changes in animals over the treatment time (for instance, every week changes).
  3. Using of mice on C57BL6 genetic background may be questionable in terms of kidney injury development, as it is pretty well known that C57BL6 mice are quite resistant to HFD-induced renal damage. So, it would be more informative to determine mesangial score based on PSR staining. H&E staining on Fig 8a seems to present glomeruli from different parts of the cortex (closer to a cortex’s apex glomeruli look smaller compared to the center part), which makes it to look different falsely. Please correct. It would also very informative to perform ORO staining to investigate if lipid droplets accumulation occurs in the HFD/clozapine group to support your earlier findings in this study. Renal iNOS is also very informative parameter to study in context of your experiments.
  4. Please explain why FABP4 and SREBP1 only were selected as fatty liver markers, while the literature reports many other genes, such as CLOCK TF, STAT3, ABCC2, PXR, etc. to be highly involved into NAFLD.
  5. I would suggest to remove Fig 1b, as this information is already included into Fig 1a.
  6. Please make shorter the paper title, 2.4 and 2.5 sub-heads.
  7. Please reduce Introduction part and Discussion part and make it more relevant to your study instead of writing a mini-review on all known data in this area of knowledge.
  8. Extensive English language editing is required.

Reviewer 3 Report

In this paper authors reported that clozapine can worsen nonalcoholic fatty liver disease, diabetes, kidney and retinal injury in a high fat diet (HFM) mice model. Higher fatty liver scores in clozapine treated mice are likely related to lowered adiponectin protein levels and increased FASN, PNPLA3 protein, FABP4 mRNA and SREBP1 mRNA levels.

The paper is well designed with a correct sequence in the illustration of the data and the discussion section is wide-ranging.

Unfortunately, this work is inappropriate. It is unoriginal.

This work is very, very similar to a recent paper of same authors. (Doxepin Exacerbates Renal Damage, Glucose Intolerance, Nonalcoholic Fatty Liver Disease, and Urinary Chromium Loss in Obese Mice.- Chang GR, Hou PH, Yang WC, Wang CM, Fan PS, Liao HJ, Chen TP.Pharmaceuticals (Basel). 2021 Mar 16;14(3):267. doi: 10.3390/ph14030267).

The exposure, the sequence of results, the phraseology adopted, the graphs and tables are exactly superimposable. Furthermore, since the drugs analyzed have the same mechanism of action, the molecular targets analyzed are exactly the same.

The research similarities are so great that they render the work unoriginal and I have no choice but to recommend rejection.

Round 2

Reviewer 2 Report

The authors have addressed all the concerns raised from the previous submission. 

Reviewer 3 Report

The new revision did not substantially change the paper.

The paper is well designed but remains substantially superimposable to the previous paper, already mentioned, on Doxepin  (doi: 10.3390/ph140302)

I am really sorry to have to reconfirm my judgment

As already reported in the first review, the feeling is "already seen". The drugs analyzed have the same mechanism of action and so the molecular targets analyzed are exactly the same. What doesn't work is the exact overlapping of the graphs and tables of the two works. The sequence of the results, the phraseology adopted. The graphs and tables are exactly superimposable. They could be interchangeable.

In my opinion the sequence of the work should be completely revised as well as the presentation of the graphs and tables so that it differs from the previous study.